# A Narrative Review on Dietary Strategies to Provide Nitric Oxide as a Non-Drug Cardiovascular Disease Therapy: Beetroot Formulations—A Smart Nutritional Intervention

**DOI:** 10.3390/foods10040859

**Published:** 2021-04-15

**Authors:** Diego dos Santos Baião, Davi Vieira Teixeira da Silva, Vania Margaret Flosi Paschoalin

**Affiliations:** Chemistry Institute, Federal University of Rio de Janeiro, Avenida Athos da Silveira Ramos 149, Cidade Universitaria, Rio de Janeiro, RJ 21941-909, Brazil; diegobaiao20@ufrj.br (D.d.S.B.); davivieira@ufrj.br (D.V.T.d.S.)

**Keywords:** green leaves, beetroot, nitrate-rich diet therapy, nitric oxide, advanced hemodynamic parameters, clinical trials

## Abstract

Beetroot is a remarkable vegetable, as its rich nitrate and bioactive compound contents ameliorate cardiovascular and metabolic functions by boosting nitric oxide synthesis and regulating gene expressions or modulating proteins and enzyme activities involved in these cellular processes. Dietary nitrate provides a physiological substrate for nitric oxide production, which promotes vasodilatation, increases blood flow and lowers blood pressure. A brief narrative and critical review on dietary nitrate intake effects are addressed herein by considering vegetable sources, dosage, intervention regimen and cardioprotective effects achieved in both healthy and cardiovascular-susceptible individuals. Compared to other nitrate-rich vegetables, beets were proven to be the best choice for non-drug therapy because of their sensorial characteristics and easy formulations that facilitate patient adherence for long periods, allied to bioaccessibility and consequent effectiveness. Beets were shown to be effective in raising nitrate and nitrite in biological fluids at levels capable of promoting sustained improvement in primary and advanced hemodynamic parameters.

## 1. Introduction

The vascular endothelium is formed by an endothelial cell monolayer that lines the interior of blood vessels, arteries and veins and cardiac chambers. This protective layer is able to generate an active antithrombotic surface by facilitating the transit of plasma and its cellular constituents throughout the vasculature, regulating the blood flow by maintaining blood vessel tone and hemostasis. The vascular endothelium can play a pivotal role in coagulation cascades and angiogenesis, intracellular signaling regulation and hormone trafficking [1]. Endothelium blood signals release autocrine and paracrine substances in response to diverse stimuli, where angiotensin II, endothelin-1, thromboxane A2 and prostacyclin H2 participate in vasoconstriction, whereas nitric oxide (NO), bradykinin, and hyperpolarizing factors contribute to vasodilation. Vascular autoregulation influences the structural integrity of vessels and circulation function and hemodynamics [2].

The endothelium-derived NO, the main vasoactive effector, is released in response to physical (sheer stress), hormonal and/or platelet-derived substances stimuli, in addition to vascular relaxation induction, platelet inhibition and leukocyte adhesion, as well as smooth muscle cell proliferation [3].

However, aging, unhealthy lifestyle and eating habits, as well as certain phyisiopathological conditions, including those grouped among the risk factors for cardiovascular disease (CVD), such as obesity, diabetes mellitus and hypercholesteremia, may lead to endothelial dysfunction development [4]. Endothelial dysfunction is considered an event that precedes atherosclerosis, as it causes an imbalance between endothelium-derived vasodilators and vasoconstrictor synthesis, resulting in diminished NO production and/or availability [5]. Nutritional and lifestyle interventions capable of restoring NO physiological levels may avoid atherogenesis reducing the risk of cardiovascular diseases, and are considered an effective intervention, mainly in populations presenting poor adherence to pharmacological therapies inherent to chronic diseases. Furthermore, nutritional interventions may also help decrease the costs of public policy healthcare through preventive clinical measures associated with CVD, relieving system burdens. Vegetables are important components of a healthy diet, since they comprise many bioactive compounds termed functional nutrients, providing benefits in the promotion and maintenance of human health [6,7].

Strong evidence suggests that nitrate (NO_3_^−^), found abundantly in leafy green vegetables, radish and beetroot, after being reduced to nitrite (NO_2_^−^), is involved in well-documented cardioprotection, since it provides a physiological substrate for NO generation through the enterosalivary NO_3_^−^-NO_2_^−^/NO pathway [8]. However, to exert beneficial human health effects, through full vascular endothelium effects, the long-term dietary intake of NO_3_^−^ at effective concentration, obtained from food or food-formulations requires popularization. Furthermore, to reach pharmacological NO concentrations, dietary NO_3_^−^ intake must be carefully planned by designing high NO_3_^−^ content formulations from NO_3_^−^-rich vegetables, in order to avoid the administration of large serving portions that are mandatory when traditional culinary formulations are used, making it difficult for individuals to adhere to proposed long-term dietary interventions [9].

In this brief narrative and critical review, the role of NO on endothelial dysfunction and how dietary NO_3_^−^ can contribute to its generation are described. Furthermore, clinical trials aiming to evaluate health benefits following the intake of dietary NO_3_^−^ from vegetables will be evaluated and compared, considering the food matrix, ingested content, NO production and consequent improvements in primary and advanced hemodynamic parameters in both healthy individuals and in individuals presenting impaired vascular function.

## 2. Nitric Oxide

NO is a low molecular weight compound produced in gaseous form, with a short-life and able to diffuse through lipid bilayers, reaching neighboring cells. Its high reactivity character is due to 11 electrons in its valence shell with an unpaired electron, allowing it to rapidly oxidize to NO_2_^−^ and NO_3_^−^ [10]. In human physiology, NO can exert antioxidant functions through its free-radical scavenger ability, thus reducing the rates of reactive oxygen species (ROS) production [11], a harmful superoxide anion (O_2_^•−^), that generates peroxynitrite (ONOO^−^), able to attack copper and iron-metalloproteins [8]. NO acts as a neurotransmitter in the central and peripheral nervous systems and is able to mediate synapse plasticity in nerve impulse transmission, by favoring the secretion of neurotransmitters or hormones in neuronal junctions [12]. In the cardiovascular system, NO modulates vascular tone by diffusing across endothelial cells, reaching vascular smooth muscle cells and, through soluble guanylate cyclase (sGC), activates the sarcoplasmic calcium (Ca^2+^) pump, decreasing intracellular Ca^2+^ content and promoting vasodilation as a result of a diminished vascular tone [11,13].

In the human body, NO is formed from the semi-essential amino acid L-arginine, generating L-citruline as a secondary product. This pathway depends on a group of enzymes, namely nitric oxide synthase (NOS), comprising the neuronal (nNOS or NOS-I) and endothelial (eNOS or NOS-III) isoforms, both constitutive and dependent on the calcium-calmodulin complex, and the inducible isoform (iNOS or NOS-II). NOS requires enzyme cofactors such as nicotinamide adenine dinucleotide phosphate (NADPH), flavin mononucleotide (FMN), flavin adenine dinucleotide (FAD) and tetrahydrobiopterin (BH4) [5]. In addition, the shear stress exerted by blood flow on endothelial cells is crucial for the activation of eNOS under physiological conditions, although other signaling molecules such as bradykinin, adenosine, vascular endothelial growth factor (VEGF), and serotonin can also lead to eNOS activation [10,14].

The aforementioned phyisiopathological conditions, i.e., aging associated to an unhealthy lifestyle and the induction of risk factors associated to cardiovascular disease (CVD), such as obesity, diabetes mellitus and hypercholesteremia, which may lead to an imbalance between the synthesis of endothelium-derived vasodilators and vasoconstrictors, may result in diminished NO production and/or availability [5]. This malevolent condition known as endothelial dysfunction precedes atherosclerosis, as the endothelium undergoes structural and functional changes that impair homeostasis and vascular tone maintenance [4]. Oxidative stress and inflammation account for the endothelial dysfunction pathogenesis [15,16]. Both physiopathological conditions reduce NO bioavailability through the action of oxidative enzymes such as NADPH oxidase, xanthine oxidase (XO), cyclooxygenases (COX), lipoxygenases (LOX), myeloperoxidases (MPO), cytochrome P450 monooxygenase and peroxidases. In addition to increased oxygen-derived free-radical production and inflammation, eNOS activity and/or expression may decrease due to metabolic impairments [15]. Under these threatening conditions, the endothelium undergoes structural and functional changes, resulting in the exposure of vascular lumen to a prothrombotic and fibrinolytic microenvironment, increasing arterial stiffness and creating favorable conditions for atherosclerosis plaque development [17].

Due to these negative phyisiopathological conditions, the search for healthy diets has been significantly emphasized worldwide [18]. Edible plant matrices contain a complex set of bioactive compounds that may act synergically, making the regular intake of fresh vegetables appealing [19,20,21]. In order to follow WHO healthy diet recommendations, the worldwide average vegetable consumption, especially in less developed countries, should be of 400 g·day^−1^ for 70 kg individuals (excluding potatoes and other starchy tubers), aiming at reducing the risk of chronic diseases in healthy and unhealthy adults (regardless of age). Among these disorders, CVD is still considered the main cause of morbidity and mortality worldwide [6]. Indeed, meta-analyses studies evaluating health-promoting nutrients have demonstrated that a high vegetable intake is one of the cornerstones of a healthy diet and is recommended to reduce the risk of development coronary heart disease and ischemia, as these food items improve cardiovascular function [22,23,24,25]. Indeed, a recent systematic review and meta-analysis calculated the summary relative risk (RR) of incidence or mortality considering a 200 g·day^−1^ intake of a combination of vegetables. The RR for coronary heart disease was of 0.84 (95% CI: 0.79–0.90, I^2^ = 61%, *n* = 15), the same as for stroke, 0.84 (95% CI: 0.76–0.92, I^2^ = 73%, *n* = 10), increasing to 0.92 (95% CI: 0.90–0.95, I^2^ = 31%, *n* = 13) for cardiovascular disease. Similar relative cardiovascular malfunction risks were observed for fruit or vegetable intakes, separately [23].

## 3. Dietary NO_3_^−^ and Endothelial Dysfunction Therapy

Until a decade ago, NO_3_^−^ was considered an unfavorable dietary-derived toxic compound, as it was wrongly associated with the development of some malignancies, such as gastric cancer. Strict standards regarding the levels of this anion were regulated in food [26]. The World Health Organization (WHO) defined an acceptable daily intake (ADI) of 3.7 mg of NO_3_^−^·kg^−1^ body weight, the same content adopted by the European Food Safety Authority. For a normal adult weighing 70 kg, this content is equivalent to ~260 mg of NO_3_^−^·day^−1^. However, vegetarian diets commonly contain >300 mg of NO_3_^−^·day^−1^ for 70 kg adults, higher than the ADI [27].

Recently, researchers have become interested in the biological NO_3_^−^ role. Findings regarding the improvement of cardiovascular function have raised a biologically plausible and widely recognized hypothesis that the NO_3_^−^ present in vegetables may serve as a physiological substrate for NO generation which, in turn, promotes vasodilation and, consequently, improves cardiovascular function [8,13].

NO_3_^−^ is a nitric acid salt, while NO_2_^−^ is a nitrous acid salt compound, formed by a single nitrogen bonded to three or two oxygen atoms, respectively. Both compounds can be obtained from endogenous and/or exogenous sources. The endogenous formation of NO_3_^−^ and NO_2_^−^ occurs through the NO metabolism via the L-arginine/NO pathway, as mentioned previously. On the other hand, the main potential exogenous source for the acquisition of NO_3_^−^ and NO_2_^−^ is through the dietary route. Through this pathway, NO is then generated by a non-enzymatic pathway from NO_2_^−^. Dietary NO_3_^−^ is reduced to NO_2_^−^ in the oral cavity by bacteria that produce the NO_3_^−^-reductase enzyme [9]. The metabolic activities of commensal bacteria species, such as *Granulicatella*, *Actinomyces*, *Veillonella*, *Prevotella*, *Neisseria*, *Haemophilus*, and *Rothia* genera that inhabit the oral cavity have a significant influence on NO_3_^−^ to NO pathway. Previous studies have shown that individuals with a higher abundance of NO_3_^−^-reducing bacteria are able to generate more salivary NO_2_^−^ and, consequently, NO, at a faster rate following dietary NO_3_^−^ ingestion [28]. However, enzymatic activity in the mouth and, consequently, the conversion of NO_3_^−^ to NO_2_^−^ may be disrupted by antibiotic use or mouthwash rinsing, since both substances inactivate bacteria cells [10]. Subsequently, NO_2_^−^ reaches the stomach and, in this acidic environment, is protonated, forming nitrous acid (HNO_2_), which decomposes non-enzymatically to NO and other bioactive nitrogen oxides such as nitrogen dioxide (NO_2_), dinitrogen trioxide (N_2_O_3_) and the nitrosonium ion (NO^+^) [9,13]. The remaining NO_3_^−^ and NO_2_^−^ in the jejunum are rapidly absorbed into the bloodstream or tissues, where their accumulation occurs in tandem with molecules endogenously synthesized by the L-arginine/NO pathway. Most NO_3_^−^ is excreted in urine, whereas a small portion is extracted by the salivary glands, concentrating this compound in the saliva, continuing the entero-salivary cycle [8,9]. A small part of plasma NO_3_^−^ and NO_2_^−^ concentrations may suffer the action of xanthine oxidoreductase (XOR), which displays similar activity to NO_3_^−^-reductase. NO_2_^−^ can also be reduced to bioactive NO by deoxyhemoglobin (deoxyHb) and deoxymyoglobin (deoxyMb), especially when O_2_ levels are low. Other enzymes and compounds exhibiting redox potential, such as aldehyde oxidase (AO), aldehyde dehydrogenase (ALDH), carbonic anhydrase (CA), vitamin C (Vit C.) and polyphenols, display the ability to synthesize NO from NO_2_^−^ reduction [8].

Several studies report beneficial effects of dietary NO_3_^−^ sources as a new physiological, therapeutic and nutritional approach to attain the intended cardioprotective effects by NO production stimulation [8,29,30]. However, dosage, supplementation regimen and individual health status must be considered to obtain the maximum cardioprotective effect following NO_3_^−^ intake. Furthermore, environmental factors such as temperature, exposure to sunlight, atmospheric humidity, water content and irradiation, as well as agricultural factors like plant genotype, fertilization, herbicide use, amount of available nitrogen, type of crop, soil conditions, nutrient availability and transport and, finally, storage conditions also influence NO_3_^−^ contents in plants, and, consequently dietary NO_3_^−^ supplementation [31].

## 4. Dietary NO_3_^−^ Vegetable Sources

Vegetables are the main source of dietary NO_3_^−^, corresponding to 85% of the daily intake, although NO_3_^−^ content can vary widely within members of distinct botanical families [32]. The NO_3_^−^ contents in plant organs can be classified from highest to lowest, as petiole ˃ leaf ˃ stem ˃ root ˃ tuber ˃ bulb ˃ fruit ˃ seed [33]. Table 1 presents a list of vegetables commonly included in Western diets considered NO_3_^−^ sources, classified according to NO_3_^−^ contents, from the highest to the lowest.

The NO_3_^−^-rich vegetables within the *Amaranthaceae* family comprise beetroot (1300 mg·kg^−1^), beet greens (1852 mg·kg^−1^), Swiss chard (1690 mg·kg^−1^), and green spinach (≈2500 mg·kg^−1^), while a *Lamiaceae* family representative consists of basil (2292 mg·kg^−1^). Concerning the *Brassicaceae* family, the most representative members are bok choy (1933 mg·kg^−1^), black radish (1271 mg·kg^−1^), turnip (1018 mg·kg^−1^), mustard greens (1160 mg·kg^−1^), rocket or arugula (4677 mg·kg^−1^), kohlrabi (1769 mg·kg^−1^), and radish (≈2000 mg·kg^−1^). *Apiaceae* family members include coriander (2445 mg·kg^−1^), celery (1100 mg·kg^−1^) and parsley (2134 mg·kg^−1^), whereas *Asteraceae* family members include lettuce (≈1800 mg·kg^−1^), leaf chicory (1452 mg·kg^−1^), and butter leaf lettuce (2000 mg·kg^−1^). All these vegetables are included in the high NO_3_^−^-containing vegetable category of > 1000 mg·kg^−1^. Vegetables such as cabbage (513 mg·kg^−1^), curly kale (987 mg·kg^−1^), broccoli (≈300 mg·kg^−1^), broccoli raab (905 mg·kg^−1^), cauliflower (202 mg·kg^−1^) and Savoy cabbage (324 mg·kg^−1^), which belong to the *Brassicaceae* family; carrot (≈300 mg·kg^−1^) and fennel (363 mg·kg^−1^), both members of the *Apiaceae* family; artichokes (174 mg·kg^−1^), asparagus chicory (355 mg·kg^−1^), and endive (663 mg·kg^−1^), belonging to the *Asteraceae* family, garlic (183 mg·kg^−1^) and green onion (≈450 mg·kg^−1^) from the *Liliaceae* family; aubergine (314 mg·kg^−1^), capsicum (108 mg·kg^−1^) and potato (220 mg·kg^−1^), belonging *Solanaceae* family; courgette (416 mg·kg^−1^) and cucumber (240 mg·kg^−1^), pumpkin (894 mg·kg^−1^), from the *Cucurbitaceae* family member all contain intermediate NO_3_^−^ concentrations ranging from 100 to 1000 mg·kg^−1^ [33,34,35,36].

Among the vegetables considered the richest dietary NO_3_^−^ sources, as listed in Table 1, beetroot, rocket and spinach have been the most tested concerning dietary interventions, and all resulted in effective improvements in cardiovascular performance estimated through blood pressure reduction and vascular function amelioration (Figure 1).

NO_3_^−^ vascular-effects depend on digestibility and bioavailability (bioacessibility), and better performances are obtained when NO_3_^−^ intake originates from food matrices compared to NaNO_3_^−^ salt administration [37]. The beneficial effects of different NO_3_^−^-rich vegetables and NO_3_^−^ doses in NO stimulation production and biochemical, hemodynamic, and vascular parameters in healthy or cardiovascular-compromised patients are summarized in Table 2. It is important to note that, to the best of our knowledge, NO_3_^−^ supplementation from green leaves has only been performed in healthy individuals, and it is unknown whether their effects can be extended to individuals presenting cardiovascular risk factors. In addition, although the cardiovascular protective effects of NO_3_^−^-enriched vegetables have been clearly demonstrated in clinical trials with healthy individuals, the large volume of juice vegetables used to achieve effective dietary NO_3_^−^ concentrations can be a limiting factor in ensuring adherence to long-term nutritional interventions. However, this NO_3_^−^ limitation does not impact supplementation by beetroot juice. Beet juice and other beetroot formulations can be ingested in comfortable serving portions to achieve threshold NO_3_^−^ concentrations in order to promote beneficial cardiovascular function effects.

A large volume of spinach comprising a serving portion of 250 g leaves containing 220 mg of NO_3_^−^ were administrated to twenty-six healthy individuals, resulting in an increase in NO synthesis evidenced by an eight-fold increase in salivary NO_2_^−^ and a seven-fold increase in salivary NO_3_^−^ at 120 min post-meal. Large artery elasticity indices were increased alongside lower pulse pressure and reduced systolic blood pressure (SBP) [46].

An amount of 800 mg NO_3_^−^ intake was supplied through four different vegetable drinks, namely beetroot juice (116 g), rocket salad (196 g), spinach (365 g) or NaNO_3_^−^ (1.1 g) prepared in water, which triggered an increase in NO_3_^−^ and NO_2_^−^ plasma concentrations. SBP declined after 150 min of beetroot juice ingestion (from 118 ± 2 to 113 ± 2 mm Hg) and a rocket salad beverage (from 122 ± 3 to 116 ± 2 mm Hg), which was sustained for at least 300 min after ingestion of the spinach beverage (from 118 ± 2 to 111 ± 3 mm Hg). Diastolic blood pressure (DBP) also declined after 150 min ingestion of all beverages and was sustained at lower levels for 300 min after rocket salad or spinach ingestion [48].

All NO_3_^−^ rich-vegetable drinks were more efficient than NaNO_3_^−^ in reducing both SBP and DBP, and beetroot was the most effective considering the food weight/NO_3_^−^ content ratio [48]. However, to the best of our knowledge, the lowest effective volume of beetroot able to promote beneficial vascular effects was 70 mL of beetroot juice containing 6.45 mmol NO_3_^−^ (403 mg), which was administered to 24 older and overweight volunteers for three weeks. This supplementation regimen and the offered dose promoted 2.3-fold and 7.3-fold increases in urinary and salivary NO_3_^−^, respectively, and resulted in a 7.3 mm Hg decrease in SBP [50].

Beetroot consumption is noteworthy as a convenient and attractive alternative to obtain cardioprotective NO_3_^−^ effects in both healthy individuals and those presenting risk factors for CVD diseases, due to the distinct but smart formulations (traditional or novel) that can be prepared to fulfill effective pharmacological dietary NO_3_^−^ concentrations. An attractive and compact NO_3_^−^-enriched-beetroot gel has been formulated in an attempt to provide an enriched NO_3_^−^ food product able to promote the claimed cardioprotective effects while still being easy to administer and facilitate adherence to nutritional therapy [44]. Acute supplementation with 100 g of beetroot gel containing 390 mg of NO_3_^−^ promoted a decrease in SBP (−6.2 mm Hg), DBP (−5.2 mm Hg), and heart rate (−7 bpm) in a pilot study conducted with healthy individuals. However, NO_3_^−^ supplementation had to be adjusted to treat hypertensive individuals, since similar doses in compromised vascular individuals do not alter hemodynamic parameters. The non-susceptibility of 27 treated hypertensive patients was clearly demonstrated by the intake of 7.0 mmol (434 mg) of NO_3_^−^ in 140 mL of beetroot juice for 7 days, resulting in increased NO synthesis, assessed by plasmatic, urinary and salivary NO_3_^−^ and NO_2_^−^, but with no differences in home and 24 h ambulatory, SBP and DBP [47]. These results indicate that, in order to ameliorate primary hemodynamic parameters, high doses of dietary NO_3_^−^ combined with a long-term intervention can be applied to treat individuals presenting impaired endothelial function. Furthermore, in an unprecedented clinical trial, patients displaying at least three risk factors for the development of CVD, including hypertension, were chronically supplemented for three weeks with an enriched NO_3_^−^ beetroot-cereal bar providing 589 mg of NO_3_^−^ in 60 g of the intervention product, resulting in 14.0 mm Hg and 6.5 mm Hg decreases in SBP and DBP, respectively, in response to ~15-fold or ~7-fold increased plasma NO_3_^−^ and NO_2_^−^ concentrations, respectively. Endothelial function in the treated volunteers was improved and arterial stiffness was reduced by 14% [45,51].

## 5. Plasma NO_3_^−^/NO_2_^−^ Increments on Cardiovascular Health and Impaired Cardiovascular Functions

It is known that plasma NO_3_^−^ and NO_2_^−^ concentrations are dependent on ingested NO_3_^−^ [52], but the minimum increase in NO_3_^−^/NO_2_^−^ plasma levels necessary to promote hemodynamic responses may differ between healthy individuals and those with compromised cardiovascular function. In a clinical trial where healthy men received dietary supplementation, 3.5-fold and 1.6-fold increases of plasma NO_3_^−^ and NO_2_^−^, respectively, resulted in significant DBP reductions and increases in endothelium-independent vasodilatation. This small but effective plasma increase was generated after the acute intake of beetroot bread (NO_3_^−^ 1.1 mmol) [39]. On the other hand, Haun et al. [49] reported plasma NO_x_ (~3-fold) and NO_2_^−^ (less than 1.5-fold) increases, albeit without any changes in hemodynamic parameters such as heart rate, DBP, SBP, FMD, radial artery pulse waves (PWA), central mean arterial pressure (CMAP) and central pulse pressure (CPP), after the acute intake of red spinach extract (NO_3_^−^ 1.45 mmol) by 15 healthy subjects. Although the dose used by Haun et al. [49] was slightly higher than by Hobbs et al. [39] trial, plasma NO_3_^−^ (>3.5-fold) and NO_2_^−^ (>1.6-fold) increases should be a determinant factor in choosing the dose required to benefit healthy populations.

In individuals with impaired cardiovascular function, the administered NO_3_^−^ dose should be able to meet two requirements: (i) promote an increase in systemic NO_3_^−^ and NO_2_^−^ higher than observed in healthy individuals; (ii) be administered in a chronic and uninterrupted manner.

Hypertensive pregnant women exhibited ~10- and ~1.5-fold increases in plasma NO_3_^−^ and NO_2_^−^, respectively, after 7 days of daily supplementation with NO_3_^−^ (6.45 mmol in beetroot juice). No significant differences were observed in plasma NO_3_^−^ and NO_2_^−^ levels measured 24 h after the initial dose, and even in the following 7-days of daily supplementation [53]. Similarly, a 1-week intake of beetroot juice (NO_3_^−^ ~6.45 mmol) in 27 treated hypertensive individuals resulted in a three-fold increase in plasma NO_3_^−^ and NO_2_^−^, with no differences in home and 24-h ambulatory blood pressures [54]. Finally, 24 overweight older subjects supplemented for 3 weeks with concentrated beet juice (~4.8–6.45 mmol) exhibited urinary NO_3_^−^ values ~3-fold higher greater than the baseline and beneficial SBP effects after juice intake. However, both urinary NO_3_^−^ and SBP returned to baseline levels 24 h after ingestion and in the first week following the end of supplementation [40]. These findings demonstrate that acute treatments able to promote systemic increases in NO_3_^−^ and NO_2_^−^ at levels similar to those observed in healthy individuals do not benefit individuals presenting cardiovascular risks.

On the other hand, clinical trials lasting more than 3 weeks or comprising higher NO_3_^−^ doses than usually applied (6–7 mmol) resulted in better hemodynamic outcomes. Hypertensive subjects treated for 4 weeks with beetroot juice (NO_3_^−^ 6.4 mmol) exhibited substantial increases in NO_3_^−^ and NO_2_^−^ plasma levels (~5.5 and ~2.7, respectively). In addition, this intervention provided sustained BP lowering of 7.7/5.2 mm Hg 24 h after the treatment, with clinical BP reduced by 7.7/2.4 mm Hg and home BP, by 8.1/3.8 mm Hg [41]. In this trial, SBP and DBP reduction peaks occurred only in the last week, highlighting the importance of a prolonged intervention.

In another trial, supplementation for 3 weeks with a high dose of dietary NO_3_^−^ concentrate in a 60 g beetroot cereal bar (9.5 mmol) resulted in ~15- and ~7-fold increases in plasma NO_3_^−^ and NO_2_^−^, respectively. This was accompanied by a considerable reduction in BP (−14.0/−6.5 mm Hg) and improvement in central hemodynamic and endothelial function parameters such as arterial stiffness, augmentation and index pressures, aortic systolic and pulse pressures and cutaneous microvascular conductance [45].

Based on these reports, individuals presenting physiopathological conditions that affect the cardiovascular system require a dietary therapy that associates high NO_3_^−^ doses capable of promoting systemic increases in NO_3_^−^ and NO_2_^−^ to levels higher than found in healthy individuals and in addition, is administered continuously (Figure 2).

In short, the aforementioned studies discussed herein suggest that frequent daily dietary NO_3_^−^ doses for long periods of time would be necessary to promote beneficial effects on blood pressure and endothelial function in populations presenting compromised vascular responsiveness. A systematic review and meta-analysis study of randomized controlled trials demonstrated that supplementation of inorganic NO_3_^−^ from beetroot juice over 14 days provoked decreases in SBP (−3.55 mm Hg; 95% CI: −4.55, −2.54 mm Hg) and DBP (−1.32 mm Hg; 95% CI: −1.97, −0.68 mm Hg). Furthermore, beneficial dietary NO_3_^−^ effects on endothelial function were associated with dose, age, and body mass index (BMI), where chronic beetroot juice supplementation improved flow-mediated dilation (FMD) and endothelium functional effects according to the administered NO_3_^−^ contents (β = 0.04, SE = 0.01, *p* < 0.001), age (β = −0.01, SE = 0.004, *p* = 0.02) and BMI (β = −0.04, SE = 0.02, *p* = 0.05) [55]. A critical review of experimental data shows that chronic dietary NO_3_^−^ ingestion is a positive vascular endothelium effector promoting vasodilatation and reducing blood pressure in compromised vascular responsiveness individuals.

However, only beetroot supplementation has been tested in acute and chronic assays in individuals with impaired cardiovascular function. Although the NO_3_^−^ content of green leaves is able to fulfill the effective NO_3_^−^ concentration in such patients, beetroot formulations may be the best non-drug strategy, since beetroot-derived formulations can concentrate the pharmacological NO_3_^−^ dosage in a small serving portion of an attractive food product, favoring continuous intake and better adherence to this nutritional intervention. This may explain the well-documented and consistent cardioprotective effects of beet products in both healthy individuals and those presenting risk factors for the development of CVD when compared with other rich-NO_3_^−^ vegetables, such as green leaves, assayed in clinical trials.

Furthermore, it is important to note that, in addition to NO_3_, vegetables are also a source of numerous phytochemicals able to increase eNOS activity in endothelial cells and contribute to NO synthesis [56,57]. Due to the great variety of polyphenols and other bioactive compounds in vegetables, it is difficult to point out individual or synergistic effects on NO generation. However, only NO_3_^−^ has been directly associated to the cardioprotective effect, since it provides the physiological substrate for NO generation via the NO_3_^−^-NO_2_^−^/NO enterosalivary pathway [8]. The administration of the same food matrix, depleted in NO_3_^−^, used as a placebo in the clinical trials had no effect on NO synthesis and hemodynamic parameters, proving that NO_3_^−^ is probably the active principle. The remaining phytochemicals after NO_3_^−^ removal, including polyphenols, which are preserved in the placebo, may promote a discrete increase in NO production but it seems they are not effective in promoting hemodynamic improvements, similar to the effect observed when NO_3_^−^ at concentrations under the pharmacological threshold is administered.

## 6. Conclusions

Vegetables are important health-promoting foods in a balanced diet, due to the presence of bioactive compounds, including dietary NO_3_^−^. Vegetables that belong to the green leaf group, such as rocket, green spinach, basil, radish, Swiss chard and bok choy, in addition to red beetroot, are considered the richest sources of dietary NO_3_^−^. Increasing dietary NO_3_^−^ ingestion results in beneficial effects in many physiological and clinical settings. Several clinical interventions with different NO_3_^−^-rich vegetables have been reported as affecting metabolic and cardiovascular functions by increasing NO concentrations and improving endothelial function by reducing BP and arterial stiffness. However, minimal or no hemodynamic and vascular beneficial effects in healthy individuals have been observed following acute NO_3_^−^ ingestion. To obtain the maximum cardioprotective effects of NO_3_^−^ intake, patient health status, as well as NO_3_^−^ dosage and supplementation regimen, must be considered.

The aforementioned studies suggest that frequent daily doses up to 6.0 mmol of dietary NO_3_^−^ for long periods of time (≥3 weeks) are required to promote beneficial blood pressure and endothelial function effects, mainly in populations with compromised vascular responsiveness such as hypertensive, metabolic syndrome, obese and older individuals.

Only beetroot supplementation has been tested in acute and chronic assays in individuals with impaired cardiovascular function. Although the NO_3_^−^ content of green leaves or other vegetables could fulfill the effective NO_3_^−^ concentration in healthy individuals, patients with impaired vascular function require a higher dose able to provide systemic increases in NO_3_^−^ and NO_2_^−^ to levels higher than those achieved in healthy individuals. Beet formulations are easier, attractive, accessible and were the only vegetable shown to be effective in promoting increased systemic NO production at the magnitude necessary to achieve the expected pharmacological effects in individuals presenting cardiovascular disease risk factors.

## Figures and Tables

**Figure 1 foods-10-00859-f001:**
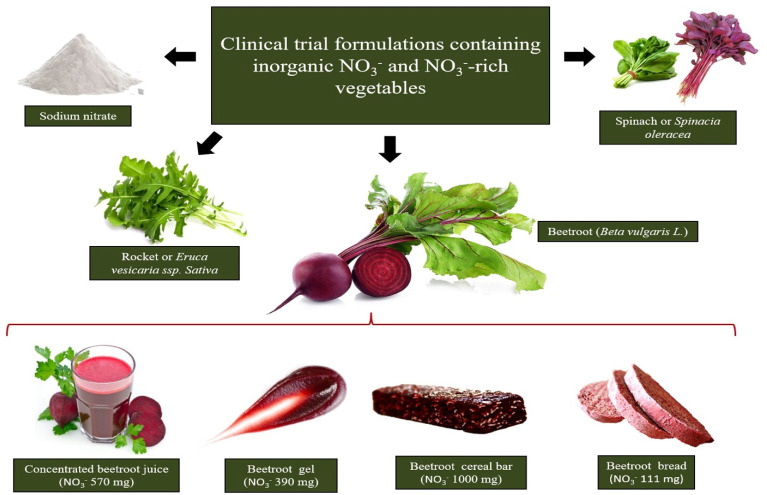
The richest sources of dietary NO_3_^−^ tested in clinical interventions are beetroot, rocket and spinach. Beetroot formulation choice to supplement dietary NO_3_^−^ relies on the design of beetroot-derived formulations containing pharmacological NO_3_^−^ doses in a small serving portion.

**Figure 2 foods-10-00859-f002:**
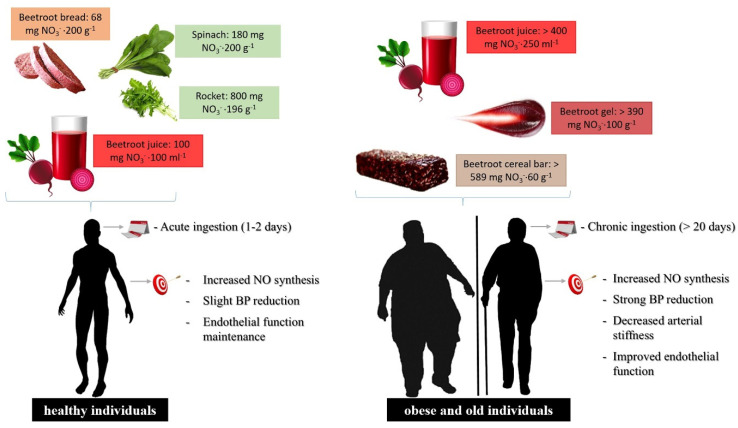
Food formulations and supplementation regimen of dietary NO_3_^−^ in healthy or cardiovascular-compromised patients. For individuals presenting risk factors for the development of cardiovascular disease, the dietary NO_3_^−^ dose should be higher to promote the systemic elevation of plasma NO_3_^−^ and NO_2_^−^ levels compared to healthy individuals, increasing NO generation by the NO_3_^−^/NO_2_^−^ pathway, where increased levels must be administered through chronic and uninterrupted supplementation.

**Table 1 foods-10-00859-t001:** Dietary NO_3_^−^ sources classified from the highest to the lowest according to mean [and range] NO_3_^−^ content.

	Vegetable	NO_3_^−^ Content/mg·kg^−1^
High NO_3_^−^ content (>1000 mg·kg^−1^)	Rocket or arugula (*Eruca vesicaria* subsp. *sativa*)	2848 [2597–3100]
	Green spinach (*Spinacia oleracea*)	2500 [2013–2797]
	Coriander (*Coriandrum sativum*)	2445
	Basil (*Ocimum basilicum*)	2292 [507–4695]
	Celery (*Apium graveolens*)	2200 [900–3500]
	Parsley (*Petroselinum crispum*)	2134 [1700–2101]
	Radish (*Raphanus raphanistrum* subsp. *sativus*)	2064 [1878–2250]
	Butter leaf lettuce (*Lactuca sativa* variety *capitata*)	2000
	Bok choy (*Brassica rapa* subsp. *chinensis*)	1933
	Lettuce (*Lactuca sativa*)	1893 [970–2782]
	Beet greens (*Beta vulgaris* subsp. *vulgaris*)	1852 [1060–2600]
	Kohlrabi (*Brassica oleracea*)	1769
	Swiss chard (*Beta vulgaris* subsp. *maritima*)	1512 [1024–2000]
	Leaf chicory (*Cichorium intybus*)	1452
	Beetroot (*Beta vulgaris* subsp. *vulgaris*)	1300 [644–1950]
	Black radish (*Raphanus raphanistrum* subsp. *sativus*)	1271 [667–1878]
	Mustard greens (*Brassica juncea*)	1160
Medium NO_3_^−^ content (100 to 1000 mg·kg^−1^)	Curly kale (*Brassica oleracea* Acephala Group)	987 [792–1181]
	Broccoli raab (*Brassica rapa*)	905
	Pumpkin (*Cucurbita pepo*)	692 [445–939]
	Turnip (*Brassica rapa* subsp. *rapa*)	684 [307–1062]
	Endive (*Cichorium endivia*)	663
	Cabbage (*Brassica oleracea* var. *capitata*)	503 [85–920]
	Green beans (*Phaseolus vulgaris*)	496 [449–585]
	Green onion (*Allium fistulosum*)	485 [99–870]
	Courgette (*Cucurbita pepo*)	416
	Fennel (*Foeniculum vulgare*)	363
	Asparagus (*Asparagus officinalis*)	355 [145–479]
	Cauliflower (*Brassica oleracea* var. *botrytis*)	331 [104–559]
	Savoy cabbage (*Brassica oleracea* Savoy Cabbage Group)	324
	Aubergine (*Solanum melongena*)	314
	Broccoli (*Brassica oleracea* var. *italica*)	300 [145–477]
	Carrot (*Daucus carota* subsp. *sativus*)	300 [121–480]
	Cucumber (*Cucumis sativus*)	240 [124–384]
	Potato (*Solanum tuberosum*)	220 [81–713]
	Garlic (*Allium sativum*)	183 [34–455]
	Artichokes (*Cynara scolymus*)	174
	Sweet pepper (*Capsicum annuum*)	117 [93–140]
	Green pepper (*Capsicum annuum*)	111 [76–159]
Low NO_3_^−^ content (<100 mg·kg^−1^)	Onion (*Allium cepa*)	87 [23–235]
	Tomato (*Solanum lycopersicum*)	69 [27–170]

NO_3_^−^ vegetables content were compiled from Lidder and Webb [8]; Hord et al. [33]; Santamaria et al. [34]; EFSA [35] and Tamme et al. [36].

**Table 2 foods-10-00859-t002:** Selected clinical trials from 2012 to 2020 compared considering administered NO_3_^−^ content, intervention duration, level of systemic increase in NO evaluated by plasma NO_3_^−^ and NO_2_^−^ levels and improvements in primary and advanced hemodynamic parameters in healthy individuals and in patients presenting impaired vascular function.

NO_3_^−^ Vegetable Intervention	NO_3_^−^ Content/Serving Portion Administered	Subjects	Duration of Administration	Trial Features	Effects	Study
White beetroot bread (*Beta vulgaris* L) Red beetroot bread (*Beta vulgaris* L)	99 mg·200 g^−1^ 112 mg·200 g^−1^	14 healthy individuals	single intake	Randomized Placebo-controlled Single-blind Crossover	↑ NO synthesis after 1 h of ingestion (through urinary NO_x_) ↓ 24 h ambulatory SBP and DBP	Hobbs et al. [38]
Beetroot bread (*Beta vulgaris* L)	68 mg·200 g^−1^	23 healthy individuals	single intake	Randomized Placebo-controlled Open-label Crossover	↑ NO synthesis after 1 h of ingestion (through plasma and urinary NO_3_^−^ and NO_2_^−^) ↓ iAUC (0–6 h after beet bread ingestion) for DBP ↑ iAUC (0–6 h after beet bread ingestion) for endothelium-independent microvascular vasodilation	Hobbs et al. [39]
Beetroot juice (*Beta vulgaris* L)	403 mg·70 mL^−1^	24 overweight older individuals	3 weeks	Randomized Placebo-controlled	↓ daily resting DBP at home	Jajja et al. [40]
	400 mg·250 mL^−1^	68 hypertensive individuals	4 weeks	Randomized Placebo-controlled Double-blind Crossover	↑ NO synthesis (by plasma and salivary NO_3_^−^, NO_2_^−^ and plasma cGMP) ↓ home, clinic and 24 h ambulatorial SBP and DBP, and arterial stiffness (through reduction of PWV and AIx) ↑ endothelial function (through increased brachial artery diameter and time to peak dilatation after FMD)	Kapil et al. [41]
	100 mg·100 mL^−1^	40 healthy individuals	single intake	Randomized Placebo-controlled Double-blind Crossover	↑ NO synthesis (by urinary NO_3_^−^ and NO_2_^−^) No significant relationships between urinary NO_3_^−^ and NO_2_^−^ concentration and body mass after intervention were observed	Baião et al. [42]
	800 mg·200 mL^−1^	14 non-hypertensive obese individuals	single intake	Randomized Placebo-controlled Crossover	↑ NO synthesis (through plasma NO_x_) ↓ ambulatory SBP following 1–6 h of moderate-intensity aerobic exercise	Bezerra et al. [43]
Beetroot gel (*Beta vulgaris* L)	390 mg·100 g^−1^	5 healthy individuals	single intake	-	↑ NO synthesis (through plasma NO_2_^−^) ↓ ambulatory SBP, DBP and HR	Silva et al. [44]
Beetroot cereal bar (*Beta vulgaris* L)	589 mg·60 g^−1^	women with 2 risk factors for CVD	3 weeks	Randomized Placebo-controlled Double-blind Crossover	↑ NO synthesis (through plasma NO_3_^−^ and NO_2_^−^) ↓ clinical DBP and SBP ↓ arterial stiffness (through reductions in AP, AIx, _ao_SP, _ao_PP, arterial age and PWV) ↑ endothelial function (through increased CVC peaks and AUC)	Baião et al. [45]
Spinach (*Spinacia oleracea*)	220 mg·250 g^−1^	26 healthy individuals	single intake	Randomized Placebo-controlled Crossover	↑ NO synthesis (through salivary NO_3_^−^ and NO_2_^−^) ↑ large artery elasticity index ↓ pulse pressure, SBP, estimated cardiac ejection time, estimated cardiac output, estimated stroke volume and total vascular impedance	Liu et al. [46]
	182 mg·200 g^−1^	30 healthy individuals	single intake	Randomized Placebo-controlled Crossover	↑ NO synthesis (through plasma RXNO, NO_2_^−^ and NO_x_) ↑ ↑ endothelial function (through increases brachial artery diameter dilatation after FMD) ↓ ambulatory SBP and pulse pressure	Bondonno et al. [47]
	800 mg·365 g^−1^	18 healthy individuals	single intake	Semi randomized Crossover	↑ NO synthesis (through plasma NO_3_^−^ and NO_2_^−^) ↓ ambulatory DBP and SBP	Jonvik et al. [48]
Red spinach (*Amaranthus dubius*)	1000 mg·90 mg^−1^	15 healthy individuals	single intake	Placebo-controlled Double-blind Crossover	↑ NO synthesis (through plasma NO_2_^−^ and NO_x_) ↑ endothelial function (through increased reactive hyperemia and calf blood flow)	Haun et al. [49]
Rocket (*Euruca vesicaria ssp. Sativa*)	800 mg·196 g^−1^	18 healthy individuals	single intake	Semi randomized Crossover	↑ NO synthesis (through plasma NO_3_^−^ and NO_2_^−^) ↓ DBP and SBP	Jonvik et al. [48]

AIx, augmentation index; _ao_SP, aortic systolic pressure; _ao_PP, aortic pulse pressure; AP, augmentation pressure; AUC, area under the perfusion curve; cGMP, cyclic guanosine monophosphate; CVC, cutaneous microvascular conductance; DBP, diastolic blood pressure; FMD, mediated flow dilatation; HR, heart rate; iAUC, incremental area under the curve; NO, nitric oxide; NO_x_, nitrate + nitrite concentration; NO_2_^−^, nitrite; NO_3_^−^, nitrate; PWV, pulse wave velocity; RXNO, S-nitrosothiols + other nitrosylated species; SBP, systolic blood pressure.

## Data Availability

Not applicable.

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
