# Peer review of "A Narrative Review on Dietary Strategies to Provide Nitric Oxide as a Non-Drug Cardiovascular Disease Therapy: Beetroot Formulations—A Smart Nutritional Intervention"

_foods, 2021, doi:10.3390/foods10040859_

Round 1

Reviewer 1 Report

First, I congratulate the authors for their work. The authors carried out an arduous and good task in the collection of recent works on the state of the topic addressed. The article provides a lot of information of interest within the scope of the journal. However, at times the text is not easy to digest for those who are not within the subject, so I recommend improving the writing to make the manuscript more enjoyable. On the other hand, the introduction is too long and makes the objective of the work diffuse. In this sense, it is necessary to be more concise and precise in the message, for which I would recommend better classifying the initial information in subsections.

Author Response

Foods

Manuscript ID: foods-1132795

Title: A narrative review on dietary strategies to provide nitric oxide as a non-drug cardiovascular disease therapy: Beetroot formulations - a smart nutritional intervention replacing the former one “A dietary strategy to provide nitric oxide as a non-drug cardiovascular disease therapy: Beetroot formulations - the smartest nutritional intervention”.

Authors: Diego dos Santos Baião, Davi Vieira Teixeira da Silva, Vania Margaret Flosi Paschoalin

GENERAL COMMENTS BY THE AUTHORS

 We believe that we have fully addressed all reviewer concerns and comments.

Several modifications were carried out in the revised manuscript: The major ones are discriminated below.

The title was changed, replacing the expression “the smartest nutritional intervention”. The introduction has been rewritten and two new topics have been added to the text according to the reviewer’s suggestions, topic 2. Nitric oxide (page 2, lines 80-142) and topic 3. Dietary NO3- and endothelial dysfunction (page 4, lines 144-199). A new table – Table 1 displaying the NO3- content of several edible plants, was included. Table 2 was modified: a new column was included in order to specify the clinical trial characteristics. Several modifications were carried out in the revised manuscript. Five news references were included:

  1. Hasler, C.M. The changing face of functional foods. J Am Coll Nutr. 2000, 19, 499S-506S. https://doi.org/10.1080/07315724.2000.10718972
  2. Mirvish, S.S. Role of N-nitroso compounds (NOC) and N-nitrosation in etiology of gastric, esophageal, nasopharyngeal and bladder cancer and contribution to cancer of known exposures to NOC. Cancer Lett. 1995, 93, 17-48. https://doi.org/10.1016/0304-3835(95)03786-V
  3. Katan, M.B. Nitrate in foods: harmful or healthy? Am J Clin Nutr. 2009, 90, 11-12. https://doi.org/10.3945/ajcn.2009.28014
  4. Ambriz-Pérez, D.L.; Leyva-López, N.; Gutierrez-Grijalva, E.P.; Heredia, J.B. Phenolic compounds: Nat-ural alternative in inflammation treatment. A Review. Cogent Food Agric. 2016, 2, 1-14. https://doi.org/10.1080/23311932.2015.1131412
  5. Appeldoorn, M.M.; Venema, D.P.; Peters, T.H.F.; Koenen, M.E.; Arts, I.C.W.; Vincken, J.P; Gruppen, H.; Keijer, J.; Hollman, P.C.H. Some Phenolic Compounds Increase the Nitric Oxide Level in Endothelial Cells in Vitro. J Agric Food Chem. 2009, 57, 7693–7699. https://doi.org/10.1021/jf901381x

All modifications were highlighted in yellow.

Modifications suggested by the reviewers have polished the manuscript and increased its overall impact. We would like to thank reviewers for his/her insights and thoughtful critiques of our manuscript. By following the reviewer`s concerns, several points in the manuscript were better addressed and discussed, improving reader understanding.

After performing the modifications suggested by the reviewers, the entire text was revised by an editing specialized company in order to improve English grammar and syntax.

Answers to Reviewer 1

Reviewer 1 comments precede our responses.

Comments and Suggestions for Authors

Comments to the Author

First, I congratulate the authors for their work. The authors carried out an arduous and good task in the collection of recent works on the state of the topic addressed. The article provides a lot of information of interest within the scope of the journal. However, at times the text is not easy to digest for those who are not within the subject, so I recommend improving the writing to make the manuscript more enjoyable. On the other hand, the introduction is too long and makes the objective of the work diffuse. In this sense, it is necessary to be more concise and precise in the message, for which I would recommend better classifying the initial information in subsections.

Answer: The introduction has been rewritten and two new topics have been added to the text to according the reviewer’s suggestions, the topic 2. Nitric oxide (page 2, lines 80-142) and the topic 3. Dietary NO3- and endothelial dysfunction (page 4, lines 144-199). Five news references were included.

Reviewer 2 Report

The manuscript provides a brief overview of the potential role of vegetable-enriched nitrate/nitrite, in particular beetroots, in the modulation of vascular/cardiovascular function.

On the whole, the topic is interesting, however it seems that other similar papers have been recently published on the general topic (Antioxidants 2020, 9, 960; doi:10.3390/antiox9100960).  Can the authors justify it and avoid duplication if the case?

Below are reported some specific comments and criticisms that should be addressed.

-Please, revise for typos (e.g. line 6, line 53), in addition some parts could benefit from an English revision.

-Title needs a revision and it should reflect more the content of the paper; i.e. a review of specific available literature on the topic. In this regard, the authors should also define the specific type of review (narrative, systematic….).

-For a better comprehension, more information should be provided on the research methodology of the literature, and the inclusion and exclusion criteria adopted for the selection of the studies.

-Lines 122-123: please revise the statement, it is not clear

-Line 132: is the paper a commentary? It seems more a narrative review

-Lines 133-135: the aim of the study should be followed by the methodologies and results while in paragraph 2 the authors report composition data.

-Lines 145-162: I would suggest to present data reported in the text as a table or figure. This would facilitate the reader.

-Line 136 and 241: I would suggest to improve the organization of the paragraphs number 2 and 3 in order to avoid repetitions and make a more clear presentation of the research findings.   

-Table 1 in this manuscript reports and discuss results present also in a table recently published by the authors

-Line 241: Authors should report and discuss that the potential beneficial effects of the beetroot formulations, spinach and rocket could derive not only from nitrate/nitrite but also from other bioactive components widely present in these food products.   

-I would suggest to revise the discussion and conclusions reducing the emphasis since the evidence from beetroots derive from six in vivo studies using different formulations (e.g. bread, juice, gel), doses, target groups and thus findings are not comparable. In addition, since vegetables are also a rich source of other bioactives (e.g. polyphenols, carotenoids, glucosinolates) the conclusions should report that the observations found could also derive from an additive/synergistic effect of the different components.

Author Response

Foods

Manuscript ID: foods-1132795

Title: A narrative review on dietary strategies to provide nitric oxide as a non-drug cardiovascular disease therapy: Beetroot formulations - a smart nutritional intervention replacing the former one “A dietary strategy to provide nitric oxide as a non-drug cardiovascular disease therapy: Beetroot formulations - the smartest nutritional intervention”.

Authors: Diego dos Santos Baião, Davi Vieira Teixeira da Silva, Vania Margaret Flosi Paschoalin

GENERAL COMMENTS BY THE AUTHORS

We believe that we have fully addressed all reviewer concerns and comments.

Several modifications were carried out in the revised manuscript: The major ones are discriminated above.

The title was changed replacing the expression “the smartest nutritional intervention”. The introduction has been rewritten and two new topics have been added to the text according to the reviewer’s suggestions, topic 2. Nitric oxide (page 2, lines 80-142) and topic 3. Dietary NO3- and endothelial dysfunction (page 4, lines 144-199). A new table – Table 1 displaying the NO3- content of several edible plants, was included. Table 2 was modified: a new column was included in order to specify the clinical trial characteristics. Several modifications were carried out in the revised manuscript. Five news references were included:

  1. Hasler, C.M. The changing face of functional foods. J Am Coll Nutr. 2000, 19, 499S-506S. https://doi.org/10.1080/07315724.2000.10718972
  2. Mirvish, S.S. Role of N-nitroso compounds (NOC) and N-nitrosation in etiology of gastric, esophageal, nasopharyngeal and bladder cancer and contribution to cancer of known exposures to NOC. Cancer Lett. 1995, 93, 17-48. https://doi.org/10.1016/0304-3835(95)03786-V
  3. Katan, M.B. Nitrate in foods: harmful or healthy? Am J Clin Nutr. 2009, 90, 11-12. https://doi.org/10.3945/ajcn.2009.28014
  4. Ambriz-Pérez, D.L.; Leyva-López, N.; Gutierrez-Grijalva, E.P.; Heredia, J.B. Phenolic compounds: Nat-ural alternative in inflammation treatment. A Review. Cogent Food Agric. 2016, 2, 1-14. https://doi.org/10.1080/23311932.2015.1131412
  5. Appeldoorn, M.M.; Venema, D.P.; Peters, T.H.F.; Koenen, M.E.; Arts, I.C.W.; Vincken, J.P; Gruppen, H.; Keijer, J.; Hollman, P.C.H. Some Phenolic Compounds Increase the Nitric Oxide Level in Endothelial Cells in Vitro. J Agric Food Chem. 2009, 57, 7693–7699. https://doi.org/10.1021/jf901381x

All modifications were highlighted in yellow.

Modifications suggested by the reviewers have polished the manuscript and increased its overall impact. We would like to thank reviewers for his/her insights and thoughtful critiques of our manuscript. By following the reviewer`s concerns, several points in the manuscript were better addressed and discussed, improving reader understanding.

After performing the modifications suggested by the reviewers, the entire text was revised by an editing specialized company in order to improve English grammar and syntax.

Answers to Reviewer 2

Reviewer 2 comments precede our responses.

Comments and Suggestions for Authors

The manuscript provides a brief overview of the potential role of vegetable-enriched nitrate/nitrite, in particular beetroots, in the modulation of vascular/cardiovascular function.

Major issues:

On the whole, the topic is interesting, however it seems that other similar papers have been recently published on the general topic (Antioxidants 2020, 9, 960; doi:10.3390/antiox9100960).  Can the authors justify it and avoid duplication if the case?

Answer: The reviewer raised a very important point. Maybe the aim of the study was not clear in the original MS or the text was not developed in the best way. To improve the MS, several modifications were made to the entire text, to clarify the importance of the conclusions on dietary NO3- therapy. All the points listed below were addressed in the revised MS:

In the aforementioned study recently published in Antioxidants, the formulation methodology and nutritional characteristics (proximate composition, nitrate and nitrite content, polyphenols, organic acids and betanin) of different beetroot formulations were highlighted. In addition, the effect of each bioactive compound present in these formulations on redox balance, gene transcription, and hemodynamic parameter modifications were reviewed. Herein, the focus is a critical review of the major sources of dietary nitrate available in vegetables rich in this compound that have already been tested in pre-clinical and clinical trials as non-drug therapies for cardiovascular diseases. The nitrate concentrations in the different vegetables and in their derived formulations, the duration of the interventions and the pharmacological outcomes in healthy and unhealthy individuals are emphasized and discussed.

Therefore, maybe the most important point in the present narrative review was to establish the best vegetable nitrate source, the ideal dose and intervention regimen to promote an increase in systemic NO3- and NO production, based on intervention trials.

Among all dietary nitrate sources, beet-derived products were shown to be effective in increasing nitrate and nitrite in biological fluids at levels capable of promoting sustained improvement in primary and advanced hemodynamic parameters. Despite not being the highest NO3- content vegetable (see Table 1, now included to the manuscript), beetroot was established as the best choice for non-drug therapy, due to its sensory characteristics, easy formulations that facilitate patient adherence for long periods of time and bioaccessibility. In other words, the most NO3- enriched vegetables are not necessarily the most suitable for dietary therapy intervention.

Below are reported some specific comments and criticisms that should be addressed.

-Please, revise for typos (e.g. line 6, line 53), in addition some parts could benefit from an English revision.

Answer: As suggested by the reviewer, the sentences in lines 7 and 44 have been corrected and the entire text has been corrected by an English reviewer.

-Title needs a revision and it should reflect more the content of the paper; i.e. a review of specific available literature on the topic. In this regard, the authors should also define the specific type of review (narrative, systematic….).

Answer: The manuscript is a narrative review. According to the reviewer's suggestion, the title of the article was changed from “A dietary strategy to provide nitric oxide as a non-drug cardiovascular disease therapy: Beetroot formulations - a smartest nutritional intervention” to “A narrative review on dietary strategies to provide nitric oxide as a non-drug cardiovascular disease therapy: Beetroot formulations - a smart nutritional intervention”.

-For a better comprehension, more information should be provided on the research methodology of the literature, and the inclusion and exclusion criteria adopted for the selection of the studies.

Answer: Table 1 was modified to include a new column describing the features of the clinical trials, specifying the time period of the trials (from 2012 to 2020) and the health benefits following the intake of dietary NO3-, sources including fresh foods or derived-formulations.

A careful revision was performed to include clear information about each trial considering the described NO3 sources and critically compared them, considering ingested content, duration of the intervention, degree of systemic increase of NO production (biological NO3- and NO2-), and improvements in primary and advanced hemodynamic parameters in healthy individuals and in those presenting impaired vascular function. As the present study is a narrative review, we did not include a methodology session as in systematic reviews and meta-analyses. After the revision, Table 2 now contains data on the population recruited in the trials including the number of individuals, sex, duration of the intervention and the studies design – included in the new added column.

-Lines 122-123: please revise the statement, it is not clear

Answer: The reviewer is correct. The sentence in page 4, lines 190-192 was changed to “Several studies report beneficial effects of dietary NO3- sources as a new physiological, therapeutic and nutritional approach to attain the intended cardioprotective effects by NO production stimulation”, for better understanding.

-Line 132: is the paper a commentary? It seems more a narrative review

Answer: The reviewer is correct, and the suggestion was accepted. The manuscript is a narrative review, so we replaced the title and modified the text for standardization.

-Lines 133-135: the aim of the study should be followed by the methodologies and results while in paragraph 2 the authors report composition data.

Answer: The manuscript sections have been reorganized so that after describing the objectives, a narrative review of the literature on the role of NO and dietary NO3- was performed, as well as a critical description of clinical trials involving dietary NO3- supplementation according to vegetable source.

-Lines 145-162: I would suggest to present data reported in the text as a table or figure. This would facilitate the reader.

Answer: We agree and thank the reviewer for the suggestion. A table (Table 1, entitled “NO3- dietary sources classified from highest to lowest according to mean [and range] of NO3- content”) was included in the manuscript (page 5) adding valuable information on the dietary sources of NO3- according to data published by Lidder & Webb [8]; Hord et al. [33]; Santamaria et al. [34]; EFSA [35] and Tamme et al. [36], standardizing the information as mg NO3-/kg vegetable.

-Line 136 and 241: I would suggest to improve the organization of the paragraphs number 2 and 3 in order to avoid repetitions and make a more clear presentation of the research findings.   

Answer: We agree with the reviewer`s suggestion. To improve the manuscript, the paragraphs now in topic “4. Dietary NO3- vegetable sources” (page 5-11, lines 201-310) were rearranged in order to avoid repetitions and clearly present the experimental/clinical findings.

-Table 1 in this manuscript reports and discuss results present also in a table recently published by the authors

Answer: In this critical review, we compare several dietary sources of beetroot, since we have been frequently questioned about the use of other nitrate sources, particularly green leaves and even NaNO3-, to supplement individuals in clinical trials. After writing the article entitled “Beetroot, a Remarkable Vegetable: Its Nitrate and Phytochemical Contents Can be Adjusted in Novel Formulations to Benefit Health and Support Cardiovascular Disease Therapies” published in Antioxidants (2020), we realized that the gap was not filled. In the Antioxidants article, Table 3 reported and discussed the health effects of human intervention trials performed in the last 5 years (2014–2019). However, we only included beetroot supplementation studies. In Table 3 – Antioxidants - we presented and discussed the supplementation regimen, biochemical and hemodynamic parameters of healthy, physically active or cardiovascular‐compromised patients in order to unite and clarify all the information available until now administering beetroot as an NO3- source. However, even after this careful compilation, we were not able to answer the questions concerning other NO3--rich vegetable supplementations, so the gap was not filled.

Therefore, we decided to compile the results from clinical trials concerning NO3-supplementation, independent of the vegetable used as the NO3- source. Herein, in Table 2, the clinical effects of dietary NO3- interventions were compared between NO3--rich vegetables (i.e., spinach and rocket). Furthermore, we discussed and compared the formulations of all vegetables used in the interventions, the supplementation regimens, and biochemical and hemodynamic parameters, including healthy and cardiovascular‐compromised patients.

We are now convinced that beetroot formulations are the best source of dietary NO3 supplementation.

Reference:

Baião, D.S.; da Silva, D.V.T.; Paschoalin, V.M.F. Beetroot, a remarkable vegetable: its nitrate and phytochemical contents can be adjusted in novel formulations to benefit health and support cardiovascular disease therapies. Antioxidants. 2020, 9, 1-36. https://doi.org/10.3390/antiox9100960

-Line 241: Authors should report and discuss that the potential beneficial effects of the beetroot formulations, spinach and rocket could derive not only from nitrate/nitrite but also from other bioactive components widely present in these food products.   

Answer: The reviewer is correct. The manuscript's title itself states A narrative review on dietary strategy strategies to provide nitric oxide as a non-drug cardiovascular disease therapy: Beetroot formulations - a smart nutritional intervention). The aim was to discuss, compare and indicate the best dietary NO3- source from vegetables able to stimulate NO production and, consequently, improve hemodynamic and vascular parameters in healthy, but, mainly, individuals presenting impaired vascular function.

At this stage, we cannot report and discuss the potential beneficial effects of the beetroot formulations, spinach and rocket from other bioactive components (such as phenolic compounds, organic acids and betalains) widely present in these food products because we had not evaluated their functional compounds. Indeed, it is an excellent idea that we will take into consideration to include in an upcoming article.

A paragraph explaining the food matrix composition on cardioprotection was included (page 14, lines 407-420) “Furthermore, it is worth mentioning that, in addition to NO3-, vegetables are sources of numerous phytochemicals able to increase the activity of eNOS in endothelial cells and, thus, contribute to NO synthesis [56,57]. Due to the great variety of polyphenols and other bioactive compounds in vegetables, it is difficult to point out individual or synergistic effects on NO generation. However only NO3- has been associated directly to cardioprotective effects, since it provides the physiological substrate for the generation of NO via the NO3- - NO2-/NO enterosalivary pathway [8]. The administration of the same food matrix-depleted in NO3- (used as a placebo in the clinical trials) has no effect on NO synthesis and hemodynamic parameters, proving that NO3- may be considered the active principle. The remaining phytochemicals, including polyphenols, preserved in the placebo, after the removal of NO3- may promote a discrete increase in NO production but it seems they are not effective in promoting hemodynamic improvements, similar to the effect observed when NO3- at concentrations under the pharmacological threshold are administered”.

-I would suggest to revise the discussion and conclusions reducing the emphasis since the evidence from beetroots derive from six in vivo studies using different formulations (e.g. bread, juice, gel), doses, target groups and thus findings are not comparable. In addition, since vegetables are also a rich source of other bioactives (e.g. polyphenols, carotenoids, glucosinolates) the conclusions should report that the observations found could also derive from an additive/synergistic effect of the different components.

Answer: The reviewer raised a very interesting point. We recognize that vegetables have other important bioactive compounds, such as phenolic compounds (Carrillo et al., 2019). Phenolic compounds have been associated with antioxidant activity against harmful reactive oxygen species, while others stimulate cellular defense mechanisms, enhancing stress responses, competing for active enzymes and receptor binding sites in subcellular structures, modulating the gene expression of proteins/enzymes capable of acting against oxi‐degenerative processes that may occur in molecules and cellular structures (Van Breda et al., 2018). However, only NO3- has been associated to cardioprotection, since it provides a physiological substrate for NO production by the enterosalivary NO3--NO2-/NO pathway (Lidder et al., 2013). This fact was demonstrated in several of the included studies, as they used NO3-free formulations (placebo) and, when placebo supplementation (vegetables with NO3- removed) was offered to volunteers, no increase in NO production or improvement in hemodynamic and vascular parameters were observed (Lara et al., 2016). This information was included in Table 2.

A sentence addressing this point raised by the reviewer was included in page 14, lines 407-420, as already mentioned in the previous response.

Reference:

Carrillo, J.A.; Zafrilla, M.P.; Marhuenda, J. Cognitive Function and Consumption of Fruit and Vegetable Poly-phenols in a Young Population: Is There a Relationship? Foods. 2019, 8, 1-17. https://doi.org/10.3390/foods8100507

Van Breda, S.G.J.; de Kok, T.M.C.M. Smart combinations of bioactive compounds in fruits and vegetables may guide new strategies for personalized prevention of chronic diseases. Mol Nutr Food Res. 2018, 62, 1700597. https://doi.org/10.1002/mnfr.201700597

Lara, J.; Ashor, A.W.; Oggioni, C.; Ahluwalia, A.; Mathers, J.C.; Siervo, M. Effects of inorganic nitrate and beetroot supplementation on endothelial function: A systematic review and meta‐analysis. Eur J Nutr. 2016, 55, 451–459. https://doi.org/10.1007/s00394-015-0872-7

Lidder, S.; Webb, A. Vascular effects of dietary nitrate (as found in green leafy vegetables and beetroot) via the nitrate‐nitrite‐nitric oxide pathway. Br J Clin Pharmacol. 2013, 75, 677–696. https://doi.org/10.1111/j.1365-2125.2012.04420.x

Reviewer 3 Report

The review work explores an old but very important aspect of world health: dietary habits and cardiovascular disease.

This review is relevant and well synthesized.

The authors have done a good compilation of interesting results and overall, the discussion is well constructed.

However, I have two suggestions.

My first suggestion is about the title, I don’t think beetroot formulations can be considered “the smartest” nutritional intervention regarding nitric oxide provision, and the authors themselves state during their comprehensive analysis that the majority of the studies have been done with few nitric oxide sources. So, in my opinion change “the smartest” to “a smart nutritional intervention” is more appropriate.

My second suggestion goes to the insertion of a table with the different dietary NO3- food sources. The paper would be highly improved with the insertion of such table as it would be easier for any researcher interested in it, localize these food sources.

Author Response

Foods

Manuscript ID: foods-1132795

Title: A narrative review on dietary strategies to provide nitric oxide as a non-drug cardiovascular disease therapy: Beetroot formulations - a smart nutritional intervention replacing the former one “A dietary strategy to provide nitric oxide as a non-drug cardiovascular disease therapy: Beetroot formulations - the smartest nutritional intervention”.

Authors: Diego dos Santos Baião, Davi Vieira Teixeira da Silva, Vania Margaret Flosi Paschoalin

GENERAL COMMENTS BY THE AUTHORS

 We believe that we have fully addressed all reviewer concerns and comments.

Several modifications were carried out in the revised manuscript: The major ones are discriminated above.

The title was changed replacing the expression “the smartest nutritional intervention”. The introduction has been rewritten and two new topics have been added to the text according to the reviewer’s suggestions, topic 2. Nitric oxide (page 2, lines 80-142) and topic 3. Dietary NO3- and endothelial dysfunction (page 4, lines 144-199). A new table – Table 1 displaying the NO3- content of several edible plants, was included. Table 2 was modified: a new column was included in order to specify the clinical trial characteristics. Several modifications were carried out in the revised manuscript. Five new references were included:

  1. Hasler, C.M. The changing face of functional foods. J Am Coll Nutr. 2000, 19, 499S-506S. https://doi.org/10.1080/07315724.2000.10718972
  2. Mirvish, S.S. Role of N-nitroso compounds (NOC) and N-nitrosation in etiology of gastric, esophageal, nasopharyngeal and bladder cancer and contribution to cancer of known exposures to NOC. Cancer Lett. 1995, 93, 17-48. https://doi.org/10.1016/0304-3835(95)03786-V
  3. Katan, M.B. Nitrate in foods: harmful or healthy? Am J Clin Nutr. 2009, 90, 11-12. https://doi.org/10.3945/ajcn.2009.28014
  4. Ambriz-Pérez, D.L.; Leyva-López, N.; Gutierrez-Grijalva, E.P.; Heredia, J.B. Phenolic compounds: Nat-ural alternative in inflammation treatment. A Review. Cogent Food Agric. 2016, 2, 1-14. https://doi.org/10.1080/23311932.2015.1131412
  5. Appeldoorn, M.M.; Venema, D.P.; Peters, T.H.F.; Koenen, M.E.; Arts, I.C.W.; Vincken, J.P; Gruppen, H.; Keijer, J.; Hollman, P.C.H. Some Phenolic Compounds Increase the Nitric Oxide Level in Endothelial Cells in Vitro. J Agric Food Chem. 2009, 57, 7693–7699. https://doi.org/10.1021/jf901381x

All modifications were highlighted in yellow.

Modifications suggested by the reviewers have polished the manuscript and increased its overall impact. We would like to thank reviewers for his/her insights and thoughtful critiques of our manuscript. By following the reviewer`s concerns, several points in the manuscript were better addressed and discussed, improving reader understanding.

After performing the modifications suggested by the reviewers, the entire text was revised by an editing specialized company in order to improve English grammar and syntax.

Answers to Reviewer 3

Reviewer 3 comments precede our responses.

Comments and Suggestions for Authors

The review work explores an old but very important aspect of world health: dietary habits and cardiovascular disease. This review is relevant and well synthesized. The authors have done a good compilation of interesting results and overall, the discussion is well constructed.

However, I have two suggestions.

Major issues:

My first suggestion is about the title, I don’t think beetroot formulations can be considered “the smartest” nutritional intervention regarding nitric oxide provision, and the authors themselves state during their comprehensive analysis that the majority of the studies have been done with few nitric oxide sources. So, in my opinion change “the smartest” to “a smart nutritional intervention” is more appropriate.

Answer: The expression “the smartest nutritional intervention” was replaced by “a smart nutritional intervention”, as suggested by reviewer 3 (page 1, lines 2-5).

My second suggestion goes to the insertion of a table with the different dietary NO3- food sources. The paper would be highly improved with the insertion of such table as it would be easier for any researcher interested in it, localize these food sources.

Answer: According to the reviewer's suggestion, a table listing the different dietary sources of NO3- classified from highest to lowest, according to mean [and range] of NO3- content, was inserted in the text. Table 2 was adapted from Lidder & Webb [8]; Hord et al. [33]; Santamaria et al. [34]; EFSA [35] and Tamme et al. [36]. NO3- contents were expressed as mg·kg-1 (page 5-6).

Round 2

Reviewer 1 Report

I congratulate the authors for the interesting work and for the substantial improvement since the first version.

Author Response

Foods

Manuscript ID: foods-1132795

Title: A narrative review on dietary strategies to provide nitric oxide as a non-drug cardiovascular disease therapy: Beetroot formulations - a smart nutritional intervention.

Authors: Diego dos Santos Baião, Davi Vieira Teixeira da Silva, Vania Margaret Flosi Paschoalin

GENERAL COMMENTS BY THE AUTHORS

 We believe we have fully addressed all reviewer 1 concerns and comments. Modifications suggested by reviewer 1 have polished the manuscript and increased its overall impact. We would like to thank reviewer 1 for his/her insights and thoughtful critiques of our manuscript. By following reviewer 1’s concerns, several points in the manuscript were better addressed and discussed, improving reader understanding.

Reviewer 2 Report

The manuscript has been greatly improved and the authors answered to most of the requests.

The manuscript still need a careful editorial reading.

As an example:

Line 279-283: long statement to be revised

Line 288: please delete concentrations

Line 312: I would suggest to avoid “unhealthy” while considering functional impairment condition as a potential substitute or similar alternatives

Line 330-333: revise the statement

Line 396: "only" is present twice

…………..  other statements need a further check

Author Response

Foods

Manuscript ID: foods-1132795

Title: A narrative review on dietary strategies to provide nitric oxide as a non-drug cardiovascular disease therapy: Beetroot formulations - a smart nutritional intervention.

Authors: Diego dos Santos Baião, Davi Vieira Teixeira da Silva, Vania Margaret Flosi Paschoalin

GENERAL COMMENTS BY THE AUTHORS

We believe that we have fully addressed all reviewer 2 concerns and comments.

Several modifications were carried out in the revised manuscript, wrong words and statement were corrected and rewritten according to reviewer 2’s suggestions.

All modifications were highlighted in yellow.

Modifications suggested by the reviewer 2 have polished the manuscript and increased its overall impact. We would like to thank reviewer 2 for his/her insights and thoughtful critiques of our manuscript. By following reviewer 2’s concerns, several points in the manuscript were better addressed and discussed, improving reader understanding.

After performing the modifications suggested by the reviewers, the entire text was revised by an editing specialized company in order to improve English grammar and syntax.

Answers to Reviewer 2 (Round 2)

Reviewer 2 comments precede our responses.

Comments and Suggestions for Authors

The manuscript has been greatly improved and the authors answered to most of the requests.

Minor issues:

The manuscript still need a careful editorial reading. As an example:

Line 279-283: long statement to be revised

Answer: As suggested by the reviewer, the statement in page 11, lines 279-284 was corrected and an English reviewer has corrected entire text.

Line 288: please delete concentrations

Answer: The statement in page 11, lines 288-289 has been rewritten repositioning the word “concentration” for better understanding.

Line 312: I would suggest to avoid “unhealthy” while considering functional impairment condition as a potential substitute or similar alternatives

Answer: The term "unhealthy" in page 11, lines 313-314 was removed as suggested by the reviewer, and replaced by impaired cardiovascular functions.

Line 330-333: revise the statement

Answer:  For better understanding, the statement in page 12, line 332-334 has been rewritten, as suggested by the reviewer.

Line 396: "only" is present twice

…………..  other statements need a further check

Answer:  The second word "only" in page 13, line 397-398 has been removed and written as follows: “However, only beetroot supplementation has been tested in acute and chronic assays in individuals with impaired cardiovascular function”. A specialized English publishing company has revised the entire text again.
